# Detection of Gas Pipeline Leakage Using Distributed Optical Fiber Sensors: Multi-Physics Analysis of Leakage-Fiber Coupling Mechanism in Soil Environment

**DOI:** 10.3390/s23125430

**Published:** 2023-06-08

**Authors:** Shuyu Zhang, Shangran Xie, Yuanzhi Li, Mengqi Yuan, Xinming Qian

**Affiliations:** 1A State Key Laboratory of Explosion Science and Technology, Beijing Institute of Technology, Beijing 100081, China; 2Yangtze Delta Region Academy, Beijing Institute of Technology, Jiaxing 314000, China; 3School of Optics and Photonics, Beijing Institute of Technology, Beijing 100081, China

**Keywords:** gas pipeline leakage monitoring, optical fiber sensor, distributed acoustic sensing, distributed temperature sensing, stress wave

## Abstract

Optical fiber sensors are newly established gas pipeline leakage monitoring technologies with advantages, including high detection sensitivity to weak leaks and suitability for harsh environments. This work presents a systematic numerical study on the multi-physics propagation and coupling process of the leakage-included stress wave to the fiber under test (FUT) through the soil layer. The results indicate that the transmitted pressure amplitude (hence the axial stress acted on FUT) and the frequency response of the transient strain signal strongly depends on the types of soil. Furthermore, it is found that soil with a higher viscous resistance is more favorable to the propagation of spherical stress waves, allowing FUT to be installed at a longer distance from the pipeline, given the sensor detection limit. By setting the detection limit of the distributed acoustic sensor to 1 nε, the feasible range between FUT and the pipeline for clay, loamy soil and silty sand is numerically determined. The gas-leakage-included temperature variation by the Joule-Thomson effect is also analyzed. Results provide a quantitative criterion on the installation condition of distributed fiber sensors buried in soil for the great-demanding gas pipeline leakage monitoring applications.

## 1. Introduction

Natural gas leakage is a critical threat to public safety in most cities worldwide [1]. For gas pipelines with medium and low gas pressure (i.e., 0–0.4 MPa), due to their vast distribution overcrowded areas and hence the irreversible damage to safety, once the explosion occurs, gas leakages induced by the aging of the pipeline draws intense research interests in the community [2]. Therefore, it is highly beneficial to establish sensitive techniques capable of detecting weak gas leaks to prevent safety hazards in the early stage.

Several branches of methods have been proposed to monitor gas pipeline leakages, including chemical composition analysis approaches [3], acoustic analysis approaches [4,5,6,7], optical measurement methods [8,9], smart-ball methods [10], and optical fiber sensors [11]. Most of the reported approaches are point sensors installed on or around the pipeline to monitor the gas leakages at one position or over a short range. Even though point sensors can achieve very high detection sensitivity and resolution, they are inappropriate for long–distance (kilometer–long) gas leakage monitoring due to their limits in the measurement range and cost factors. On the other hand, optical fiber sensors can monitor the gas leakages over the entire pipeline based on various light backscattering effects along FUT [12]. Since commercial available optical fibers can guide light with an extremely low loss (~0.2 dB/km), distributed optical fiber sensors (DOFS) are therefore possible to guard the pipeline with tens–of–kilometer lengths using one single sensing fiber and one signal demodulator, providing a cost–effective solution for pipeline monitoring. DOFSs can probe the local temperature and strain variations along the fiber based on the backscattered Rayleigh or Raman scattering effect [13]. In addition, DOFSs possess high measurement sensitivities due to the involved light interference process hence are beneficial for detecting small leaks along the pipeline. DOFSs also can survive in harsh (humid, corrosiveness, high temperature, high pressure) environments due to the chemical stability of fuse silica, drawing great attention in recent years in the field of structural health monitoring [14], pipeline monitoring [15], etc.

As early as 1989, Abe discussed sensors that used twisted fibers to detect the strain distribution along the length of the optical cables [16]. compared the difference between DOFSs and strain gauges, demonstrating the advantages of DOFSs in long–range strain measurements [17]. Wang et al. used distributed optical fiber temperature sensors to monitor the leakage of natural gas pipelines and found that monitoring was limited by spatial resolution [18]. Utilized fiber Bragg grating and long–period fiber gratings to assess the feasibility of a non–destructive assessment of infrastructure using Portland cement concrete and asphalt mixtures for temperature, strain, and level monitoring. They found that the fiber optic sensor measurements were consistent with the simulation results [19]. When leakage occurs in a buried natural gas pipeline, the signal collected by the optical fiber sensor is affected by the stress wave propagation through the soil, involving a rather complicated fluid–structure interaction process [20]. Even though DOFSs have been applied in varieties of scenarios, systematic analysis of the stress wave propagation and temperature distribution in the soil induced by the gas leakage, as well as the coupling mechanism between the soil and the fiber, are still rare at this stage, especially for gas pipelines with medium and low gas pressure. In this work, we report a detailed investigation of the leakage–fiber coupling mechanism, including soil’s pressure wave and heat propagation processes. As a result, the ultimately applied strain on FUT can be obtained. Furthermore, the dependence of the detected strain and temperature distribution on key parameters such as pipeline–fiber distance, leakage port diameter, and type of soils are revealed.

## 2. Principle of Distributed Optical Fiber Strain and Temperature Sensors

DOFSs adopt the principle of optical time domain reflectometer (OTDR). In this scheme, laser pulses are launched into FUT, and backscattered light generated in the fiber (Rayleigh, Raman, or Brillouin scattering) is collected at the same end of the laser source. The backscattered light’s properties (intensity, frequency, phase, or polarization) are changed by the measurands (strain or temperature), which can then be detected and demodulated to retrieve the information along the fiber. The laser pulse’s time–of–flight corresponds to the position over the fiber [21,22,23,24,25].

The general scheme of OTDR–based DOFSs is sketched in Figure 1. Lightwave generated from a narrow–linewidth laser source (at 1550 nm wavelength) is firstly modulated by an acousto–optic modulator (AOM) to form a pulse train, which is then amplified by an erbium–doped fiber amplifier (EDFA). Laser pulses are sent into FUT through an optical circulator, and the backscattered light is collected by a photodiode (PD), converting the optical signal to the electronic one. The signal is then digitized by an analog–to–digital converter (ADC) and is finally demodulated on a personal computer (PC) [26,27].

In the distributed acoustic sensor (DAS) case, Rayleigh backscattered light generated within the pulse train along FUT is collected [28]. Due to the long coherent length of the laser source, the collected backscattered light can interfere with each other making it sensitive to the local strain change. In addition, external acoustic wave introduces local strain variation on the fiber, further varying the optical path length and then the detected interference signal. By applying a proper demodulation algorithm, the waveform of the acoustic wave can be accurately retrieved [28]. DAS sensors can respond to the local strain variation with a sensitivity of nε [29]. In the distributed temperature sensor (DTS), Raman backscattered light is collected. The intensity ratio between anti–Stokes and Stokes components is measured, correlating with the local temperature change along the fiber [30].

## 3. Analytical Model: Spherical Stress Wave Propagation

The propagation of gas leakage from the pipeline in the soil can be modelled by a spherical stress wave propagation process [31]. In this model, the soil is considered as a standard linear solid (see Figure 2) with its stress *σ* expressed as:(1)$$\sigma ={Ε}_{a}\varepsilon +{Ε}_{M}\int_{0}^{t}\dot{\varepsilon }e^{-\frac{t-\tau }{\theta_{M}}}d\tau$$
where *E_a_* is elastic constants of equivalent linear elastic elements, *ε* is strain, *E_M_* is the elastic modulus of the Maxwell bulk element (Wang, 2005), $\dot{\varepsilon }$ and is strain rate. *θ_M_* = *η_M_*/*E_M_* is the relaxation time of Maxwell’s body, where *η_M_* is the viscosity coefficient, *τ* is the relaxation time, *t* is the time frame. With these definitions, the bulk modulus, instantaneous shear modulus, and shear modulus of the elastic element of soil can be calculated as *K* = (*E_a_* + *E_M_*)/(3 − 6*μ*), *G* = (*E_a_* + *E_M_*)/(2 + 2*μ*), and *G_a_* = *E_a_*/(2 + 2*μ*), respectively, where *μ* is Poisson’s ratio. The Maxwell body elastic and shear modulus is then given by *E* = *E_a_
*+ *E_M_* and *G_M_
*= *G* − *G_a_*.

Based on the linear viscoelastic Zhu–Wang–Tang (ZWT) constitutive relation [32], the solution of the spherical stress wave equation read as [31]:(2)$$u_{\infty }(r)=-\frac{\varphi_{\infty }(r)}{r^{2}}=\frac{\sigma_{0s}r_{0}^{3}}{4G_{e}r^{2}}$$
(3)$$\varepsilon_{r\infty }(r)=\frac{2\varphi_{\infty }(r)}{r^{3}}=-\frac{\sigma_{0s}r_{0}^{3}}{2G_{e}r^{3}}$$
(4)$$\varepsilon_{\theta \infty }(r)=-\frac{\varphi_{\infty }(r)}{r^{3}}=\frac{\sigma_{0s}r_{0}^{3}}{4G_{e}r^{3}}$$
(5)$$\varepsilon_{r\infty }(r)=4G\frac{\varphi_{\infty }(r)}{r^{3}}=-\frac{\sigma_{0s}r_{0}^{3}}{r^{3}}$$
(6)$$\varepsilon_{\theta \infty }(r)=-2G_{e}\frac{\varphi_{\infty }(r)}{r^{3}}=\frac{\sigma_{0s}r_{0}^{3}}{2r^{3}}$$
(7)$${\dot{\varepsilon }}_{r\infty }(r)=0,\;{\dot{\varepsilon }}_{\theta \infty }(r)=0$$
(8)$$\gamma_{\infty }(r)=0$$
where *u* is the radial displacement, *φ* is the reduced displacement potential, *r* is the radial coordinate, *r*_0_ is the distance from the boundary of the spherical cavity, *G_e_* is the shear modulus. The subscripts *r* and *θ* are the radial and tangential components, respectively. According to Equations (2)–(8), the amplitude of the radial stress of the propagating viscoelastic stress wave can be written as:(9)$$\sigma_{r}=\frac{\sigma_{r\max}r_{0}e^{-\alpha (r-r_{0})}}{r}$$
where *σ_r_*_max_ is the peak value of the radial stress, and *α* is the constitutive attenuation coefficient of soil. It shows that the attenuation of the stress wave in the soil is the combined effect of the geometric attenuation of spherical waves (1/*r* term) and the constitutive attenuation of linear viscoelastic material ($e^{-\alpha (r-r_{0})}$ term).

## 4. Numerical Model

Figure 3 illustrates the schematic diagram of the considered system. The blue tube represents the gas pipe on which a circular leakage port with diameter *d* is present. FUT is illustrated by the thin, dark blue line with its distance to the leakage port *D*. The pipe and fiber are considered *L* = 2 m in the analysis. The shallow grey box represents the area filled by soil. Natural gas (with orange dots illustrating the gas molecules) flowed from the right to left–the hand side. The leakage–induced spherical stress wave (dashed lines) is detected by the optical fiber. In the simulation, the pipeline diameter was set as 63 mm. The pipe wall thickness was neglected in the simulation since it barely affected the results. The leakage hole was put in the middle of the pipe, with its diameter *d* varying from 2 mm to 15 mm. The distance *D* was changed from 0.1 m to 0.9 m to test the response of FUT to the leakage.

### 4.1. Simulation Procedure

Figure 4 shows the flow chart of the numerical simulation. Fluent and Transient structural modules in the ANSYS Workbench were used to study the fluid–structure interaction. In Fluent, after selecting the steady–state calculation of the gas leakage, the results calculated by Fluent (Solution) were imported into the setup of Transient structural, with the data flow imported into the transient mechanics section. In the Transient structural module, both ends of the fiber were set as remote displacement constraints. The propagated pressure value on the fiber surface was used to simulate the induced strain along the fiber. FUT was regarded as formed by fused silica with its relevant physical parameters listed in Table 1.

### 4.2. Verification of Grid Independence

The grid size of Fluent mesh was determined to guarantee the accuracy of the simulation. Figure 5 plots the simulated pressure on the fiber surface versus the grid size. It can be seen that the pressure value starts to converge when the grid size is smaller than 20 mm. In the simulation, *d* = 2 mm, *D* = 0.5 m, and the soil area was clay. Table 2 lists the mesh grid parameters for the tested 6 cases in Figure 5. It can be seen that the degree of distortion (indicated by the mesh quality) remains unchanged for all cases. Therefore, a grid size of 20 mm (case 4), corresponding to the grid number 220,054, was selected for the calculation.

## 5. Results and Discussion

### 5.1. Gas Leakage Simulation

Firstly, the pressure and velocity fields induced by gas leakage in the pipeline are simulated in Fluent. The soil was considered a porous medium, and three soil types (clay, loam, and silty sand) were simulated, the physical parameters listed in Table 3. In the simulation, the pressure in the pipeline was 0.1 MPa, and the leakage port diameter was set as *d* = 2 mm. The leakage hole was set as the pressure inlet. The turbulence intensity was unity along with a 2–mm–wide hydraulic diameter. The top surface was set as the pressure outlet, the turbulence intensity of which was set to 2 and the hydraulic diameter was calculated as 666.7 mm. Other surfaces were considered wall surfaces.

Figure 6 plots the distribution of pressure (top panel) and particle velocity (bottom panel) fields near the leakage hole within the soil domain for the three soil types. It can be seen that the soil type can affect both the spatial distribution of the pressure and velocity fields and their absolute values. For pressure waves, clay has the lowest propagation loss among the considered soils, with a minimum value of 3.4 Pa after 0.5 m propagation. Silty sand, due to its smallest viscous and inertia drag coefficients, results in the largest propagation loss of the pressure wave with a minimum detectable pressure of only 0.4 Pa. Interestingly, on the other hand, the field of particle velocity shows an opposite trend, namely, the particle oscillation induced by the leaking gas attenuates the most in clay soil, reducing to 0.5 m/s after 0.5 m propagation. Silty sand shows the smallest velocity attenuation (reducing to 51.5 m/s). The results indicate that the magnitude of the pressure field is inversely proportional to the magnitude of the viscosity resistance and inertial resistance of the soil, and the magnitude of the velocity field is proportional to the magnitude of the soil viscosity resistance and inertial resistance.

### 5.2. Leakage–Induced Static Strain on FUT

The pressure wave propagated over the soil environment results in an axial strain component along the fiber, which can be conveniently picked up by a DAS system. The static strain experienced at the optical fiber can be used to identify the maximum distance between FUT and the pipeline, given the minimum detectable strain value of DAS. The data points in Figure 7 plot the simulated axial strain along the fiber versus the distance *D* between FUT and the leakage port for clay, loam soil, and silty sand. In the simulation, a monitoring point was set at the midpoint of FUT with *D* varying from 0.1 m to 0.9 m. The displayed strain values are the static ones in the equilibrium state. As the distance between the optical fiber and the leakage port increases, as expected, the detected strain by the optical fiber decreases. The dashed–black horizontal line indicates the DAS sensor’s detection limit (1 nε). It can be seen that for the same *D*, clay soil offers the largest strain value on the fiber. In the case of clay soil, *D* can be set as far as 0.7 m, in which case the axial strain can still be detected by DAS. For loamy soil, the maximum *D* reduces to 0.6 m, while for silty sand, it reduces to less than 0.1 m. This can be explained by the smallest constitutive attenuation coefficient of stress wave propagation in clay soil among the three types of considered soils. For optical fibers, the relation between stress σ_f_ and strain *ε_f_* satisfies:(10)$$E_{f}=\frac{\sigma_{f}}{\varepsilon_{f}}$$
where *E_f_* is the elastic modulus of the optical fiber. The linear relation between *σ_f_* and *ε_f_* suggests that the numerically simulated axial strain data can also be fitted by Equation (10). The solid lines in Figure 7 plot the fit of the simulated data to Equation (10), showing an excellent agreement with the model. This indicates that the spherical stress wave model in Section 3 is suitable for our analysis. The fitted values of *α* are listed in the first column of Table 4.

Since DAS can monitor the strain distribution over the fiber length, we plot the simulated strain distribution for the three soil types, as shown in Figure 8a–c. In the simulation, the diameter of the leakage port *d* was set as 2 mm, 5 mm, and 15 mm to investigate the capability of weak leakage detection by using a DAS sensor. It can be seen that when *d* = 2 mm, for clay soil, the sensor can properly resolve the strain distribution when *D* < 0.5 m, while for loamy soil, the distance *D* can only be less than 0.1 m. In silty sand, the leakage cannot be detected by DAS (not shown) except that the fiber is attached to the pipeline due to the high propagation loss of the stress wave. When *d* = 5 mm, it can be seen from Figure 8b that DAS can monitor the strain with a maximum distance of *D*~0.7 m in clay, and this distance increases to 0.9 m when *d* = 15 mm (Figure 8c). The observed distortions in the shape of the strain distribution in Figure 8b,c are induced by the unstable speed and pressure transmitted to the surface of the fiber due to the relatively large leakage hole and the close distance to the fiber.

### 5.3. Leakage–Induced Transient Strain on FUT

The gas leakage generates an abrupt stress wave in the soil and, thus, a transient strain change on the fiber. In this section, we investigate the dynamic response of the DAS system to the leakage–induced stress wave. In the simulation, the parameters were set as *D* = 0.1 m, *L* = 2 m and *d* = 15 mm. The sampling frequency of the DAS system was set as 1 kHz. Clay (Figure 9a), loamy soil (Figure 9b) and silty sand (Figure 9c) were considered. The top panel plots the transient evolution of the axial strain acting on the fiber over time. For all types of soil, the leakage first induces an obvious temporal oscillation of the strain, and then the oscillation damps and converges to a static value shown in Section 5.2. However, the oscillation amplitude and frequency range of the observed strain oscillation is significantly distinct for different types of soil. This can be more clearly seen in the middle panel of Figure 9a–c, which plots the power spectrum of the fiber–detected strain signal by performing a Fourier transform on the time domain signal. For all types of soil, the main peak of the strain variation is around 10 Hz, with other peaks showing the strong harmonics of the fundamental frequency.

On the other hand, it can be seen that clay has the largest energy of the fundamental oscillation component due to the relatively weak attenuation of the stress wave. In contrast, silty sand has an amplitude of more than 50 dB weaker than clay due to the stronger stress wave attenuation, agreeing with the observation in Section 5.1. To further identify the feature of the resolvable strain signal, a continuous wavelet transform (bottom panel) was performed on the time domain signal, showing the evolution of the instantaneous frequency of the strain over time. For loamy soil and silty sand, some high-frequency components around 0.25 kHz appear while it quickly decays (in less than 50 ms), and for clay, the oscillation last for a longer time, indicating that it is easier for the DAS sensor to identify the leakage when the fiber is buried in clay soil.

### 5.4. Leakage–Induced Temperature Variation along FUT

The gas leakage in the pipeline can also induce temperature variation along the pipeline due to the throttling effect (also known as the Joule–Thomson effect), a phenomenon causing the gas temperature to drop sharply in the vicinity of the leakage port since the gas pressure is abruptly reduced [33,34,35]. In this case, DOFSs, especially DTS based on Raman–scattering effect, may be used to determine the position of the gas leakage by monitoring the temperature variation along the pipeline [36]. However, DTSs mainly monitor gas leakage in medium and high-pressure pipelines (2.5–4.0 MPa) [36,37]. Here we study the possibility of using DTS to monitor the leakage of low–pressure gas pipelines.

The Joule–Thomson coefficient *C_J_* is defined as the rate of temperature change over time during an isenthalpic process, or the amount of temperature change caused by a reduction in unit pressure [34,38,39]:(11)$$C_{J}=\underset{\Delta T\to 0}{\lim}{\left(\frac{\partial T}{\partial P}\right)}_{H}$$
where *T* is the gas temperature in the pipeline, *P* is the gas pressure in the pipeline, *H* is enthalpy. In a low–pressure gas pipeline, the weak gas leakage can be regarded as an adiabatic throttling expansion process. CJT is generally positive, indicating a drop in temperature around the leakage port and hence the surrounding soil and the fiber. The absolute temperature variation Δ*T* of the leaked gas can be obtained as:(12)$$\Delta T=C_{J}\cdot \Delta P$$
where Δ*P* is the difference between the gas pressure in the pipeline and ambient pressure. Equation (6) suggests that when Δ*P* = 0.1 MPa (in the case of low–pressure pipeline), Δ*T*~0.42 K around the leakage port. To investigate the effect of the soil on heat propagation, a numerical model was established, as shown in Figure 10, in which the length of the pipeline and the fiber *L* = 1 m, the diameter of leakage port *d* = 2 mm, and the distance *D* = 0.05 m. The gas pressure inside and outside the pipeline is 0.2 MPa and 0.1 MPa, respectively (thus Δ*P* = 0.1 MPa), and the initial temperature of the system was set at 300 K.

Figure 10a plots the simulated temperature distribution along the fiber when the fiber is attached to the pipeline (i.e., *D* = 0 mm), showing that at the leakage port, the temperature is lower than the surrounding environment. The simulated temperature difference is around 0.68 K, agreeing with the result predicted by Equation (6). This amount of temperature variation is within the measurement sensitivity of the DTS sensor (~0.1 °C [40]. After propagation over 0.05 m in clay soil, however, our simulation shows that the observed temperature variation on FUT reduces to less than 0.01 °C, mainly due to the soil’s huge value of heat capacity. This suggests that to monitor the leakage of low–pressure gas pipelines, FUT must be attached to the pipeline. Note that in reality, the errors in the measured leakage parameters (e.g., the strain and temperature variation as well as the retrieved leakage amplitude and the pressure) by the DAS system can be estimated through a repeat measurement under the same experimental condition and further calculating the standard deviation from the average values.

## 6. Conclusions

We have investigated the multi–physics coupling process between the leakage–induced stress wave and the sensing fiber. It is identified that a larger value of soil viscosity resistance is more beneficial for the propagation of leakage–induced stress waves, leading to higher sensitivity for the DAS system. In the case of clay soil, when the internal pressure of the pipeline is 0.1 MPa, the maximum laying distance between the sensing fiber and the pipeline is 0.9 m in the case 15 mm leakage hole diameter, while it reduces to 0.5 m when the leakage hole diameter is 2 mm, indicating that the maximum laying distance should not exceed 0.5 m for weak leakage detection. Among the soil types, it is also revealed that clay has the largest strain power spectrum amplitude and the longest duration of leakage strain oscillation, resulting in a longer detection window for the leakage. Finally, the results indicate that the DTS sensor can be used for leakage detection of low–pressure gas pipelines (0–0.4 MPa). The sensing fiber needs to be attached to the pipeline to ensure that the observed temperature changes are within the measurement sensitivity range of DTS.

## Figures and Tables

**Figure 1 sensors-23-05430-f001:**
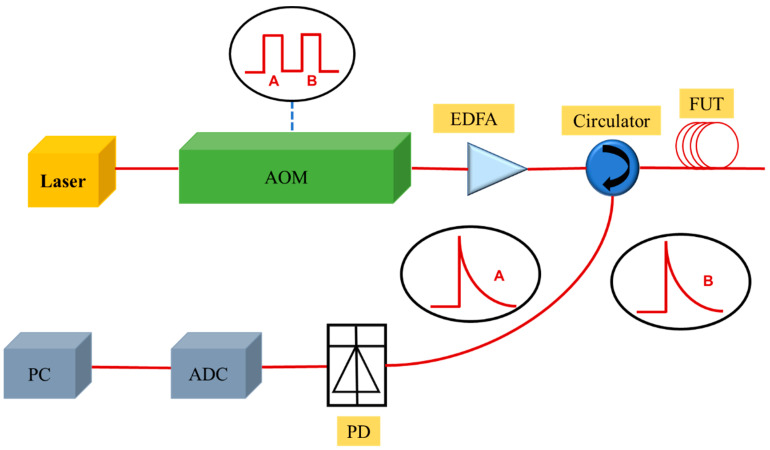
Schematic diagram of a generally distributed optical fiber sensor. AOM, acousto–optic modulator; EDFA, erbium–doped fiber amplifier; FUT, fiber under test; PD, photodiode; ADC, analog–to–digital converter; PC, personal computer.

**Figure 2 sensors-23-05430-f002:**
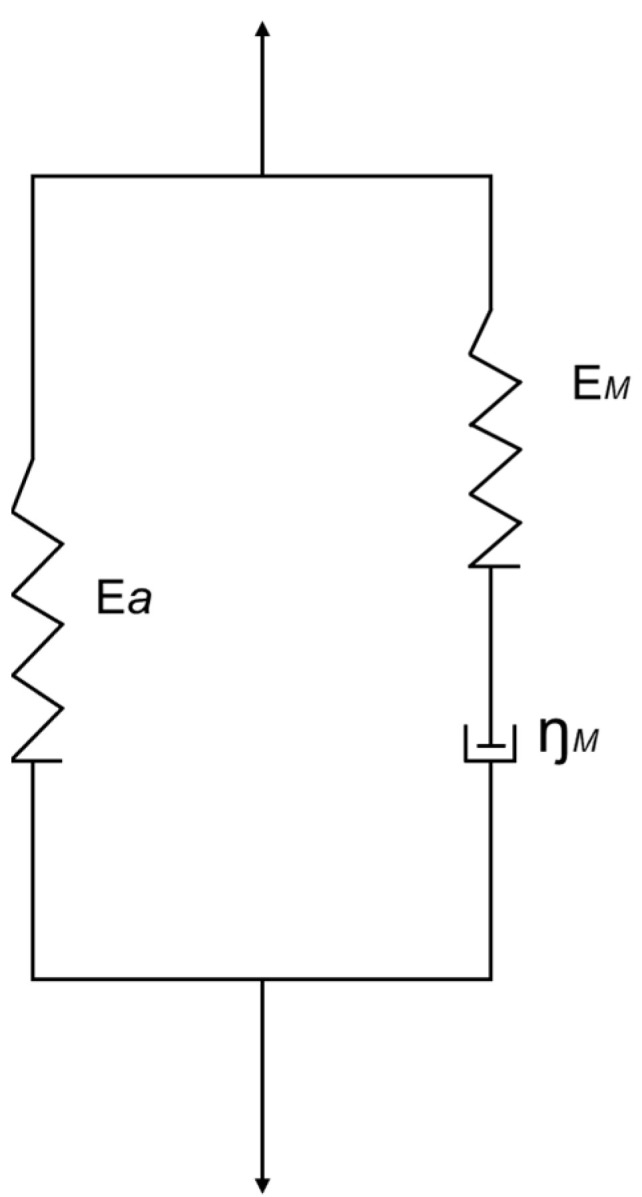
Generalized linear solid model for the soil.

**Figure 3 sensors-23-05430-f003:**
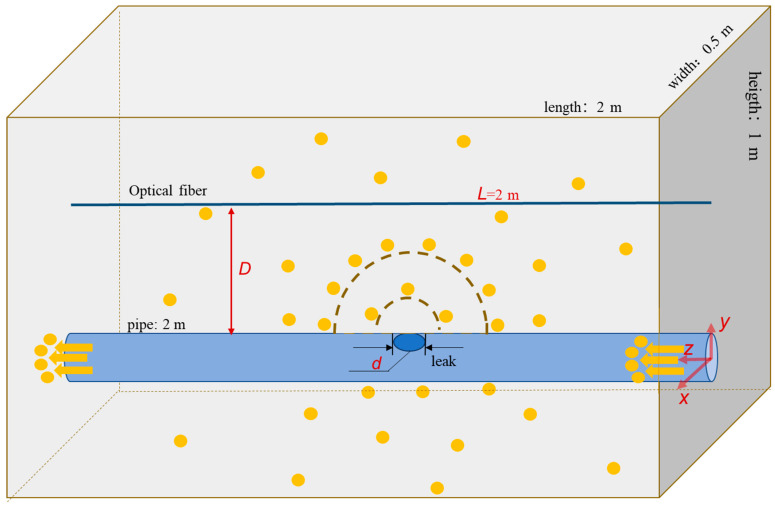
Sketch of the considered system.

**Figure 4 sensors-23-05430-f004:**
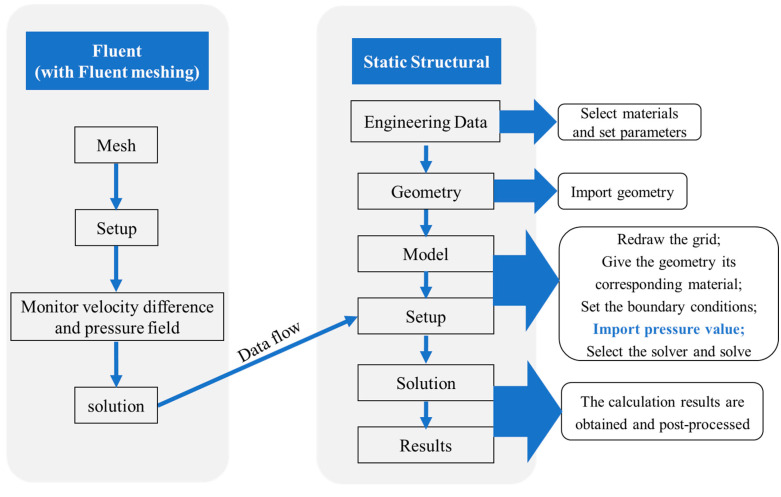
Flow chart of the numerical simulation.

**Figure 5 sensors-23-05430-f005:**
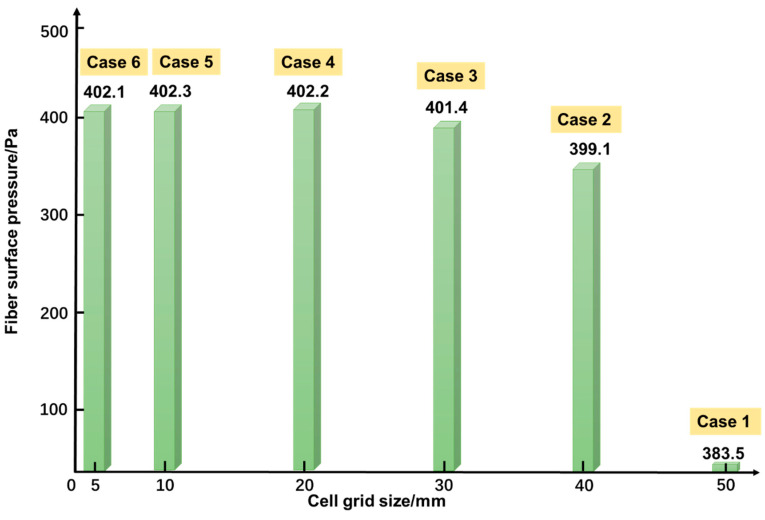
Simulated pressure on the fiber surface versus cell grid size.

**Figure 6 sensors-23-05430-f006:**
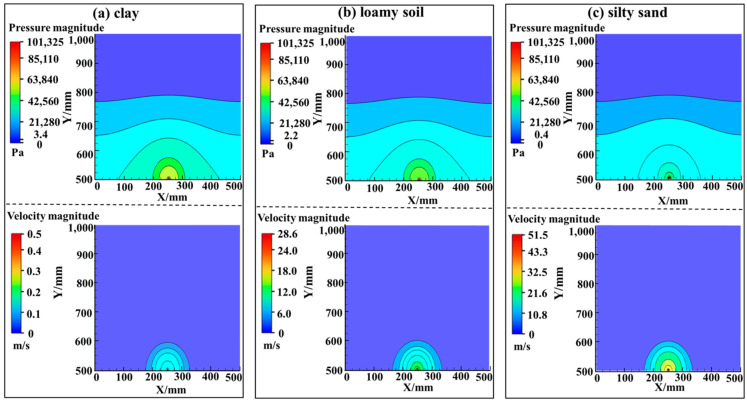
Distribution of pressure and velocity field in the vicinity of the pipeline leakage port buried in (**a**) clay soil, (**b**) loamy soil and (**c**) silty sand.

**Figure 7 sensors-23-05430-f007:**
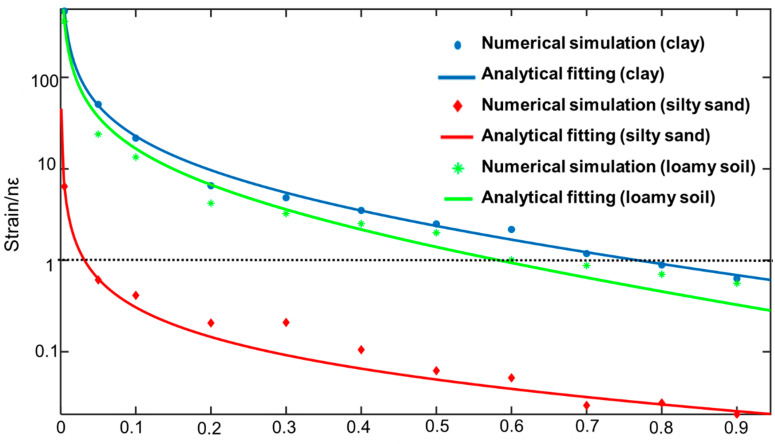
The strain curve of three soil types versus the distance between FUT and the leakage hole *D* (*d* = 2 mm).

**Figure 8 sensors-23-05430-f008:**
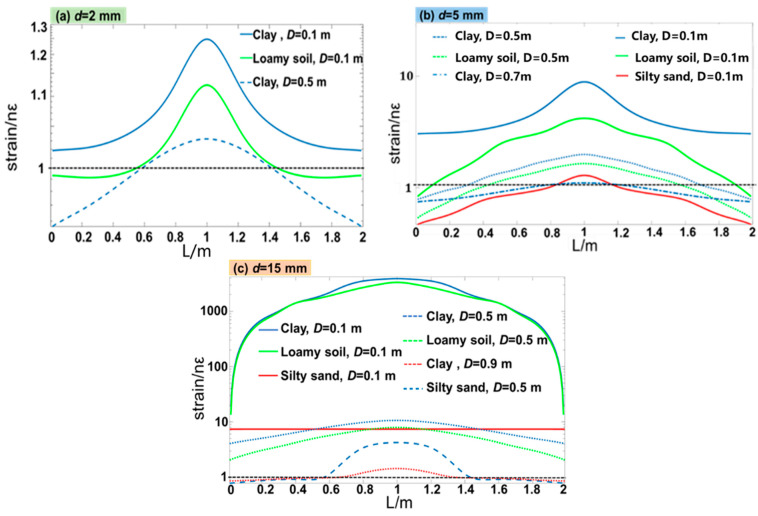
The variation strain along FUT for three types of soil when (**a**) *d* = 2 mm, (**b**) *d* = 5 mm, and (**c**) *d* = 15 mm.

**Figure 9 sensors-23-05430-f009:**
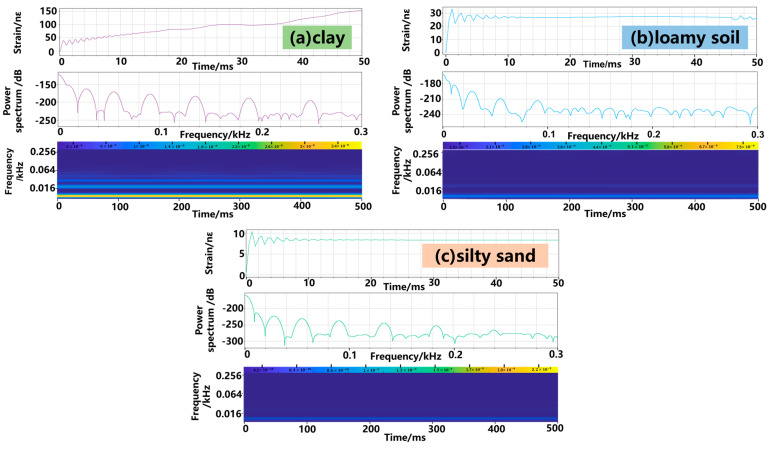
Transient strain response acted on FUT for (**a**) clay, (**b**) loamy soil, and (**c**) silty sand.

**Figure 10 sensors-23-05430-f010:**
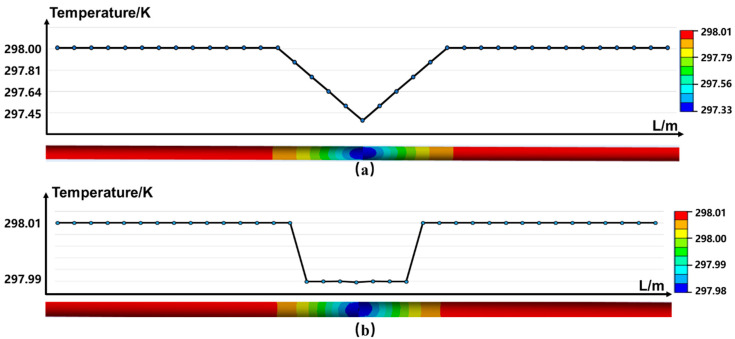
Temperature distribution along FUT after propagating in the clay (**a**) *D* = 0 mm (**b**) *D* = 0.05 mm.

**Table 1 sensors-23-05430-t001:** List of physical parameters of the optical sensing fiber.

	Density (kg/m^3^)	Young’s Modulus (GPa)	Poisson’s Ratio	Bulk Modulus (Pa)	Shear Modulus (Pa)
Value	2220	73	0.17	3.6869 × 10^10^	3.1197 × 10^10^

**Table 2 sensors-23-05430-t002:** List of mesh grid parameters for tested cases.

Case	Number of Grids	Mesh Quality	Pressure on Fiber Surface (Pa)	Grid Cell Size/mm
1	63,036	0.2	385.5	50
2	163,552	0.2	399.1	40
3	182,008	0.2	401.4	30
4	220,054	0.2	402.2	20
5	319,240	0.2	402.3	10
6	336,523	0.2	402.1	5

**Table 3 sensors-23-05430-t003:** List of physical parameters of the considered types of soil.

Soil Type	Average Particle Diameter (mm)	Porosity (%)	Viscous Drag Coefficient (m^–2^)	Inertia Drag Coefficient (m^–1^)
Silty sand	0.50	25	2.16 × 10^10^	3.36 × 10^5^
loam	0.05	43	2.45 × 10^11^	5.02 × 10^5^
clay	0.01	30	2.72 × 10^13^	9.07 × 10^6^

**Table 4 sensors-23-05430-t004:** List fitting parameters to Equation (3) for the three soil types.

Type of Soil	*α* (m^–1^)
clay	1.151
loamy soil	2.168
silty sand	3.919

## Data Availability

Data sharing is not applicable to this article as no new data were created or analyzed in this study.

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
