# Peer review of "Detection of Gas Pipeline Leakage Using Distributed Optical Fiber Sensors: Multi-Physics Analysis of Leakage-Fiber Coupling Mechanism in Soil Environment"

_sensors, 2023, doi:10.3390/s23125430_

Round 1
Reviewer 1 Report
The authors reported a detailed multi-physics analysis of the leakage-fiber coupling mechanism based on DAS system. The work identifies several key factors related with the acoustic wave propagation effects in soil as well as its coupling with sensing fibers, which can greatly influence the sensitivity of the DAS-based leakage sensor. The results are convincedly supported by the systematic analytical and numerical simulations, and the paper is very well written. The reviewer therefore warmly recommends the publish of this work in Sensors, given the following minor issues well addressed.
1. Fig. 9 is a bit difficult to understand. What is the physical meaning of the shaded area in the bottom panel? Is it possible to remove the shaded area?
2. It is suggested to remove the number “297.9998” in Fig. 10(b), since the resolution seems unrealistic for experiments.
3. How to perform the error analysis in case of this DAS-based leakage sensor?
4. The conclusion section can be further simplified.
Reviewer 2 Report
In this paper, the authors investigate the multi-physics coupling process between the leakage-induced stress wave and the sensing fibre. They find that the spherical stress wave propagation model can be used to describe the problem, with clay being the most favourable soil type for Distributed Acoustic Sensing (DAS) applications. Additionally, they show that Distributed Temperature Sensing (DTS) sensors can be used to detect leakage in low-pressure gas pipelines, provided the sensing fibre is attached to the pipeline to ensure temperature variation is within the sensor's measurement sensitivity.
I believe that the submitted version of the article deserves to be published with minor corrections:
1. In the given context, FUT stands for "fiber under test," but the authors refer to it as "the sensing fiber" being used in the experiment. It would be more clear to use an appropriate acronym.
2. Could the authors provide more details about the fibre used for their estimations? Additionally, it would be helpful to know if different types of fibres have been considered by the authors when estimating the sensing capabilities.
3. I would recommend redrawing Figure 6 in a more appropriate and informative style, including necessary axes and presenting it as a 2D plot, as is customary.
The presented simulation results are of high quality, and I hope the authors will provide experimental confirmation of their outcomes in future work.
The quality of the English language in this manuscript is understandable and appropriate for academic-style writing.
